# Genetic Profiles of Aggressive Variants of Papillary Thyroid Carcinomas

**DOI:** 10.3390/cancers13040892

**Published:** 2021-02-20

**Authors:** Meihua Jin, Dong Eun Song, Jonghwa Ahn, Eyun Song, Yu-Mi Lee, Tae-Yon Sung, Tae Yong Kim, Won Bae Kim, Young Kee Shong, Min Ji Jeon, Won Gu Kim

**Affiliations:** 1Division of Endocrinology and Metabolism, Department of Internal Medicine, Asan Medical Center, University of Ulsan College of Medicine, 88, Olympic-ro 43-gil, Songpa-gu, Seoul 05505, Korea; meihua711@naver.com (M.J.); happyjh88@gmail.com (J.A.); tykim@amc.seoul.kr (T.Y.K.); kimwb@amc.seoul.kr (W.B.K.); ykshong@amc.seoul.kr (Y.K.S.); 2Department of Pathology, Asan Medical Center, University of Ulsan College of Medicine, 88, Olympic-ro 43-gil, Songpa-gu, Seoul 05505, Korea; hipuha@hanmail.net; 3Division of Endocrinology and Metablosim, Department of Internal Medicine, Korea University College of Medicine and School of Medicine, Seoul 08308, Korea; eyunsong@gmail.com; 4Department of Surgery, Asan Medical Center, University of Ulsan College of Medicine, 88, Olympic-ro 43-gil, Songpa-gu, Seoul 05505, Korea; niphredil@amc.seoul.kr (Y.-M.L.); tysung@amc.seoul.kr (T.-Y.S.)

**Keywords:** genetic profile, next-generation sequencing, aggressive variants, papillary thyroid carcinoma, tall cell variant, columnar cell variant

## Abstract

**Simple Summary:**

Aggressive variants of papillary thyroid carcinoma (PTC) are associated with unfavorable clinical outcomes. However, limited data exist on the genetic profile of these variants of PTC. We performed targeted next-generation sequencing in 36 tissue samples from patients with aggressive variants of PTC. Aggressive variants of PTC had a higher prevalence of the *BRAF* mutation and a lower prevalence of *RAS* mutation than other types of thyroid cancer. The prevalence of mutations in the *TERT* promoter, *TP53*, and genes encoding histone methyl transferases (HMTs), switch/sucrose non-fermenting (SWI/SNF) chromatin remodeling complex, and the phosphoinositide 3-kinase/protein kinase B (PKB/AKT)/mammalian target of the rapamycin (PI3K/AKT/mTOR) pathway was between the range of PTCs and poorly differentiated/anaplastic carcinoma from The Cancer Genome Atlas (TCGA) and the Memorial Sloan Kettering Cancer Center (MSKCC) data.

**Abstract:**

Aggressive variants of papillary thyroid carcinoma (PTC) have been described with increasing frequency and are associated with unfavorable clinical outcomes. However, limited data exist on the comprehensive genetic profile of these variants. We performed targeted next-generation sequencing in 36 patients with aggressive variants of PTC and compared it to PTC from The Cancer Genome Atlas (TCGA) project and poorly differentiated thyroid cancers (PDTCs)/anaplastic thyroid cancers (ATCs) from the Memorial Sloan Kettering Cancer Center (MSKCC). *BRAF* mutation was the most prevalent (89%) in aggressive variants of PTC compared to that in other thyroid cancers. *RAS* mutation was identified in one patient (3%), which was less frequent than in others. *TERT* promoter mutation (17%) ranged between that of PTCs (9%) and PDTCs (40%). Tumor suppressor genes, *ZFHX3, TP53*, and *CHEK2*, were mutated in 14%, 3%, and 6% of aggressive variants of PTC, respectively. The mutation rate of *TP53* (3%) was significantly higher than that of PTCs (0.7%) and lower than that of ATCs (73%). Mutations in three functional groups, histone methyl transferases, SWI/SNF chromatin remodeling complex, and the PI3K/AKT/mTOR pathway, were present in 11%, 14%, and 11% of samples, respectively. In conclusion, aggressive variants of PTC had higher *BRAF* and lower *NRAS* mutation prevalence than other thyroid cancers. The prevalence of mutations in the *TERT* promoter, *TP53*, and genes encoding three functional groups ranged between that of PTCs and PDTCs/ATCs.

## 1. Introduction

Histological variants of well-differentiated papillary carcinoma (PTC) are well known, including tall cell, columnar cell, diffuse sclerosing, follicular, and hobnail variants [1,2]. Some of these variants are reported to present more aggressive behavior than classic PTC [3,4,5]. The ICCR (International Collaboration on Cancer Reporting) database reports that some of the variants of PTC including tall cell variant (TCV), diffuse sclerosing variant, and cribriform–morular variant appear to exhibit aggressive behaviors [6]. On the other hand, the current American Thyroid Association (ATA) guidelines define aggressive variants of PTC as the following three subtypes: TCV, columnal cell variant (CCV), and hobnail variant (HV) [7]. These variants were considered as a risk factor of recurrent or persistent disease; hence, total thyroidectomy and subsequent radioactive iodine (RAI) therapy were suggested as treatment options [7].

In 2014, The Cancer Genome Atlas (TCGA) project reported a comprehensive investigation of the genomic landscape of PTC, which included 496 patients. However, most (90.6%) were classical type and follicular variant (FV) PTC, and only 35 (7.5%) patients with TCV-PTC were evaluated. Additionally, the Memorial Sloan Kettering Cancer Center (MSKCC) group reported extensive cancer gene exome sequencing in 84 patients with poorly differentiated thyroid cancer (PDTC) and 33 patients with anaplastic thyroid cancer (ATC) in 2016 [8].

Despite notable progress in the elucidation of genetic profiles of thyroid cancers, studies investigating the mutational profile of aggressive variant PTC are limited and unclear. The *BRAF^V600E^* mutation, which has emerged as a marker of aggressive behavior in thyroid cancer, was observed in 80–100% of TCV, 33% of CCV, and 25.3–57.1% of HV [9,10,11,12,13]. The *RAS* mutation, which may play an important role in follicular thyroid carcinogenesis, is more frequent and may be more relevant as a prognostic indicator in follicular pattern lesions (FV-PTC, follicular thyroid carcinoma (FTC), and PDTC) than in classical PTC [14,15]. In a study evaluating 12 patients with TCV-PTC, no mutation was identified in *NRAS* and *HRAS* [16]. Other genes including the telomerase reverse transcriptase (*TERT*) promoter, tumor suppressor genes, and genes regulating key pathways and functional groups including histone methyl transferases (HMTs), switch/sucrose non-fermenting (SWI/SNF) chromatin remodeling complex, and phosphoinositide 3-kinase/protein kinase B(PKB/AKT)/mammalian target of the rapamycin (PI3K/AKT/mTOR) pathway are not well investigated in aggressive variants of PTC.

As aggressive variants of PTC are frequently identified and have been associated with poor clinical outcomes, elucidation of the mutational profile of these tumors is critical. Furthermore, comparison of the genetic profiles between different thyroid carcinomas is necessary as the clinical behavior and prognosis of the aggressive variants of PTC range between that of classical type PTC and PDTC or ATC. Hence, this study aimed to investigate the genetic profiles of aggressive variants of PTC using targeted next-generation sequencing (NGS) and compare them with those of PTCs and with those of PDTC and ATC from TCGA and MSKCC data, respectively.

## 2. Materials and Methods

### 2.1. Patients and Tissue Samples

We collected 43 tissue samples with aggressive variants of PTC between 2018 and 2020, at the Asan Medical Center, Seoul, Korea. Seven tissue samples were excluded as targeted NGS was not performed accurately. In total, tissue samples from 36 patients were finally included. Tissue samples from paired normal thyroid were not available in the current study. All the patients underwent thyroid surgery between 2004 and 2018 and were diagnosed with either tall cell variant PTC (TCV-PTC) or columnal cell variant PTC (CCV-PTC). An experienced endocrinology pathologist (D.E.S.) reviewed all specimens and selected adequate tissue blocks for DNA isolation. All subjects gave their informed consent for inclusion before they participated in the study. The study was conducted in accordance with the Declaration of Helsinki, and the protocol was approved by the Ethics Committee of Institutional Review Board of the Asan Medical Center (IRB No. 2018-0430).

### 2.2. DNA Extraction and Targeted NGS

DNA extraction and the targeted NGS process were conducted as previously described [17]. Targeted NGS was performed for 50 genes (Appendix A) using the Hiseq^TM^2500 platform (Illumina Inc., San Diego, CA, USA). Briefly, genomic DNA extracted from 2–5 μm thick sections from each block was fragmented by adaptive focused acoustic technology (Covaris Inc., Woburn, MA, USA). We hybridized 250 ng of the DNA library with SureSelect exome capture baits for exome capture. The standard exome capture libraries were prepared using the Agilent SureSelect Target Enrichment protocol (Agilent Technologies, version B.3, Illumina Inc, San Diego, CA, USA). After amplification and purification, the product was quantified according to the qPCR Quantification Protocol Guide (Illumina, San Diego, CA, USA) and qualified using TapeStation DNA ScreenTape (Agilent Technologies) [18].

### 2.3. Analysis Process

The sequenced reads were aligned and analyzed using the human reference genome (National Center for Biotechology information [NCBI] build 37) with the Burrows–Wheeler aligner (version 0.7.12, Sourceforge, San Diego, CA, USA ), the Picard tools (version 1.130, Broad Institute, Cambridege, MA, USA), and the Genome Analysis Toolkit (GATK version 3.4.0, Broad Institute, Cambridege, MA, USA) [17]. Variant genotyping was performed with Haplotype Caller of GATK, and these variants were annotated by SnpEff (version 4.1g, SnpEff & SnpSift Documentation), dbSNP (version 142, National Center for Biotechnology information, Bethesda, MD, USA), the 1000 genome project (phase 3), ClinVar, and ESP6500 [17]. Common germline variants from the somatic variant candidates and false-positive variants were manually filtered out. For further analysis, we compared the genetic profiles of frequently altered driver mutations, *TERT* promotor mutations, and other related mutations in aggressive variants of PTC with those of PTC from The Cancer Genome Atlas (TCGA) data, as well as those of poor differentiated thyroid carcinoma (PDTC) and anaplastic thyroid carcinoma (ATC) from Memorial Sloan Kettering Cancer Center (MSKCC) data analyzed by the MSK-IMPACT panel [8,18]. In addition, we compared the genetic profiles to 526 advanced PTC patients from four recent studies [17,19,20,21]. Genes encoding three functional groups including HMTs, SWI/SNF chromatin remodeling complex, and the PI3K/AKT/mTOR pathway were evaluated between different types of thyroid cancers [8,20].

### 2.4. Statistical Analysis

Statistical analyses were conducted using R (version 3.5.1, R Foundation for Statistical Computing, Vienna, Austria; http://www.R.project.org, accessed on 1 November 2020). Continuous variables are presented as the median and interquartile range (IQR), and categorical variables are presented as percentages. We used the chi-squared test to compare categorical variables. A *p*-value < 0.05 was considered statistically significant.

## 3. Results

### 3.1. Baseline Characteristics of Patients with Aggressive Variants of PTC

Table 1 presents the baseline characteristics of the patients with aggressive variants of PTC included in this study. The median age of the patients was 43.5 years (interquartile range (IQR) 34.8–51.0), with 75% of the patients being female. A total of 36 patients comprised 25 (69.4%) TCV-PTC and 11 (30.6%) CCV-PTC. A majority of patients (94.4%) underwent total thyroidectomy, while 18 patients (50.0%) underwent a modified radical neck dissection (MRND). The median primary tumor size was 2.0 cm (IQR 1.2–2.8) and 47.2% tumors were larger than 2 cm. Extrathyroidal extension (ETE) was present in 29 patients (82.9%), while 19 patients (52.8%) showed gross ETE. Cervical lymph node (LN) metastasis was identified in 32 patients (88.9%) and 18 (50.0%) had N1b disease. None of the patients had synchronous distant metastasis. Thirty-one patients (86.1%) were stage I according to American Joint Committee on Cancer tumor–node–metastasis (AJCC TNM), eighth edition. From the date of surgery to that of analysis (Nov. 2020), the median follow-up duration was 2.4 years, and seven patients (19.4%) had disease recurrence after initial treatment.

### 3.2. Somatic Mutations in Aggressive Variants of PTC

Targeted NGS results of 36 primary thyroid tumors are shown in Figure 1. The total mutation rate of each patient was shown in Figure 1A. *BRAF* mutation showed the highest prevalence (89%) of the identified variants, followed by *TERT* promoter mutation (17%). There was no difference in mutation profiles between TCV-PTC and CCV-PTC (Appendix A).

### 3.3. Mutatios in Drivers and Other Relevant Genes

*BRAF* mutation was present in 31 samples (89%), i.e., 23 (92.0%) TCV-PTC and eight (72.7%) CCV-PTC (Figure 1B). The prevalence of *BRAF* mutation was higher than that in other types of thyroid cancers (Figure 2). The prevalence of *BRAF* mutation was 59% in TCGA data, 33% in PDTCs, and 46% in ATCs in MSKCC data (Figure 2A–D). The prevalence was also higher than that in advanced PTCs (73%) from other studies (Figure 2E). *RAS* mutation was found in one patient (3%) among aggressive variants of PTC (Figure 1C), which was less frequent than other mutations (Figure 2). Both *BRAF* and *NRAS* mutations in TCV-PTC from current study were similar to those in TCV-PTC from TCGA (Appendix A).

### 3.4. Mutations in TERT Promoter

The *TERT* promoter mutation was present in 17% of samples (Figure 1D). As shown in Figure 2, the prevalence of *TERT* promoter mutation in aggressive variants of PTC was higher than that of PTCs from TCGA (9%), whereas it was lower than that of advanced PTCs (60%), as well as PDTCs (40%) and ATCs (73%) from MSKCC data.

### 3.5. Mutations in Tumor Suppressor Genes

*ZFHX3, TP53*, and *CHEK2* were mutated in 14%, 3%, and 6% of aggressive variants of PTCs, respectively (Figure 1E). Somatic mutations in other tumor suppressor genes including *ATM, RB1, NF2,* and *MEN1* were not found in the current study. The mutational prevalence of *TP53* was 3% in aggressive variants of PTC, which is considered a hallmark of advanced thyroid carcinoma; this value was higher than that of PTCs from TCGA (0.7%), but lower than that of advanced PTCs (10%), as well as PDTCs (8%) or ATCs (10%) from MSKCC data (Figure 2).

### 3.6. Mutations in Key Pathways and Functional Groups

Genetic profiles encoding three functional groups (HMTs, SWI/SNF chromatin remodeling complex, and the PI3K/AKT/mTOR pathway) are shown in Figure 1F. *KMT2C*, one of the HMT genes, was present in 11% of aggressive variants of PTC (Table 2). Genes encoding SWI/SNF chromatin remodeling complex were altered in 14% of samples, 6% in *ARID1B*, 6% in *ARID2*, and 3% in *ATRX*. No mutation was noted in *ARID5B*, *SMARCB1*, and *PBRM1*. The prevalence of mutations in HMTs (11%) and SWI/SNF chromatin remodeling complex (14%) in aggressive variants of PTC ranged between that of PDTCs (7% and 24%) and ATCs (6% and 36%) from MSKCC data (Figure 3A–D). Mutations of genes encoding the members of the PI3K/AKT/mTOR pathway were found in 11% of samples: 3% in *AKT1*, 3% in *AKT3*, and 6% in *ERRB3*. The alteration of all functional groups was 34% in aggressive variants of PTCs, 20% in TCGA, and 61% in ATCs from MSKCC data (Figure 3).

## 4. Discussion

Papillary thyroid carcinoma is the most common type of thyroid cancer, and the majority of PTCs have an excellent prognosis with a 10 year survival rate of more than 90% [22,23]. However, several uncommon histological variants of PTC, including TCV, CCV, and HV have more aggressive clinical courses, such as higher rates of recurrence, distant metastasis, and resistance to RAI therapy [3,5,24,25]. Among 36 patients with aggressive variants of PTC enrolled in this study, seven (19.4%) patients had disease recurrence within 3 years. Increased attention to the clinical importance of aggressive variants of PTC warrants investigation of the genetic profiles of these variants of PTC.

In this study, we performed targeted NGS of aggressive variants of PTC in 36 patients and compared the results with that of PTC from TCGA project, and PDTC and ATC from the MSKCC group. *BRAF* mutation showed the highest prevalence (89%) in aggressive variants of PTC, which was higher than that in PDTCs and ATCs from MSKCC data. *RAS* mutation was found in only one patient (3%), which was less frequent than other thyroid cancers. *TERT* promoter mutation (17%) ranged between that of PTCs from TCGA and PDTCs and ATCs from MSKCC data. The prevalence of *TP53* was higher than that of PTCs and lower than that of ATCs. Furthermore, the prevalence of mutations in genes encoding three functional groups (HMTs, SWI/SNF chromatin remodeling complex, and the PI3K/AKT/mTOR pathway) in aggressive variants ranged between that of PTCs from TCGA data and PDTCs and ATCs from MSKCC data. These genetic profiles explained the clinical outcomes of aggressive variants of PTCs, which were poorer than classical PTCs and better than PDTCs or ATCs.

The *BRAF^V600E^* mutation is the most common genetic alteration in PTC, and it is associated with poor clinical outcomes such as regional recurrence and distant metastasis [26,27,28]. The results of the current study are consistent with those of previous studies, which reported that *BRAF* mutation was detected in 80–100% of TCV-PTC and 33% in CCV-PTC [2,9]. As most (69.4%) of the tissue sample from current study was TCV-PTC, the frequency of *BRAF* mutation was 89%, which was close to that of TCV-PTC. In TCGA project, which predominantly includes the classical type and follicular variant PTC (90.6%), the prevalence of *BRAF* mutation was 59% [18], lower than that detected in the present study. *RAS* mutation plays an important role in early events of follicular thyroid carcinogenesis, hence; it is more prevalent in follicular patterned cancers than in classical PTC [15]. The prevalence of *RAS* mutation was previously reported as 10% in PTC, 25–30% in FTC (25–30%), 55% in PDTC, and 52% in ATC [29]. However, few studies investigated *RAS* mutation in aggressive variants of PTC [16]. In the present study, *RAS* mutation was detected in only one patient (3%) with CCV-PTC. TCV-PTC (*n* = 25) samples did not exhibit *RAS* mutation, which was consistent with the result of TCV-PTC (*n* = 29) from TCGA project (Appendix A). This may be explained by the assumption that *RAS* mutations can be predisposed to differentiation loss in thyroid cancer; thus, it is less frequent in aggressive variants of PTC [26,30]. Additionally, this result may be related to the small sample size of TCV-PTCs. Therefore, large, multicenter studies are required to establish the incidence of *RAS* mutation in aggressive variants of PTC.

The prevalence of *TERT* promoter mutation increases in the order of differentiated thyroid carcinoma, PDTCs, and ATCs [26,31,32]. In the present study, *TERT* promoter mutation was detected in 17% of samples, which was between PTCs from TCGA (9%) and PDTCs (40%) and ATCs (73%) from MSKCC data. In PTC, *TERT* promoter mutation was more frequent in *BRAF* mutated tumors [31], which is congruent with the results of this study. Of the six patients with *TERT* promoter mutations, five (83.3%) had a coexisting *BRAF* mutation. *TP53,* a tumor suppressor gene, exhibited 0.7% prevalence in TCGA study and 73% in ATCs in MSKCC data. A previous study that evaluated 18 patients with TCV-PTC showed that 61% exhibited the *TP53* mutation [33], which was significantly higher than that in the present study (3%). Although both studies used a small sample size, there was a considerable difference in frequency; hence, further studies are warranted in different centers.

The present study has several limitations. First, a small sample size of patients with an aggressive variant of PTC were evaluated, and tissue samples of HV-PTC were not available. Second, we performed targeted sequencing with only 50 thyroid-cancer-related genes. Thus, the total accumulation rate of functional genes may be underestimated when compared to other data. Third, this study lacked data on *RET* or *ALK* rearrangements and some of the mutation frequencies reported in the present study might be overestimated due to the lack of paired normal sequencing. Fourth, we also evaluated the mutational profiles of 526 advanced PTC patients from four recent studies. These also included subsets of aggressive variants of PTCs. However, the majority of patients were from the study of Pozdeyev et al. [19] and the proportion of aggressive variants of PTC was unknown in this study. These advanced PTCs consisted of clinically advanced PTCs and those with distant metastasis or mortality cases. Therefore, a larger size for further investigation into aggressive variants of PTC is required to circumvent the current research limitations. However, the present study represents the first comprehensive analysis of the genetic profile of aggressive variants of PTC, as well as its comparison to that of TCGA and MSKCC groups.

## 5. Conclusions

In conclusion, aggressive variants of PTC had a higher prevalence of *BRAF* mutation (89%) and lower prevalence of *RAS* mutation (3%) than other thyroid cancers. The mutations in *TERT* promoter (17%), *TP53* (3%), and genes encoding three functional groups (HMTs, SWI/SNF chromatin remodeling complex, and the PI3K/AKT/mTOR pathway) ranged between that of PTCs from TCGA and PDTCs and ATCs from the MSKCC group. These genetic profiles might be associated with the clinical outcomes of aggressive variants of PTC.

## Figures and Tables

**Figure 1 cancers-13-00892-f001:**
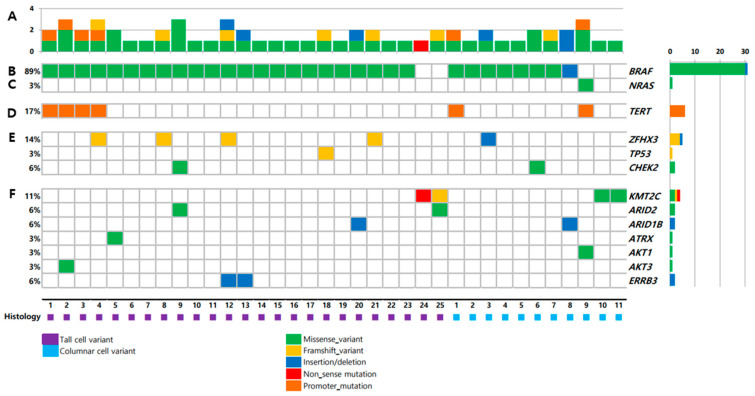
Mutations in tumors from 36 patients: (**A**) number of mutations; (**B**) *BRAF*; (**C**) *NRAS*; (**D**) *TERT* promoter; (**E**) tumor suppressor genes; (**F**) key pathways and functional groups.

**Figure 2 cancers-13-00892-f002:**
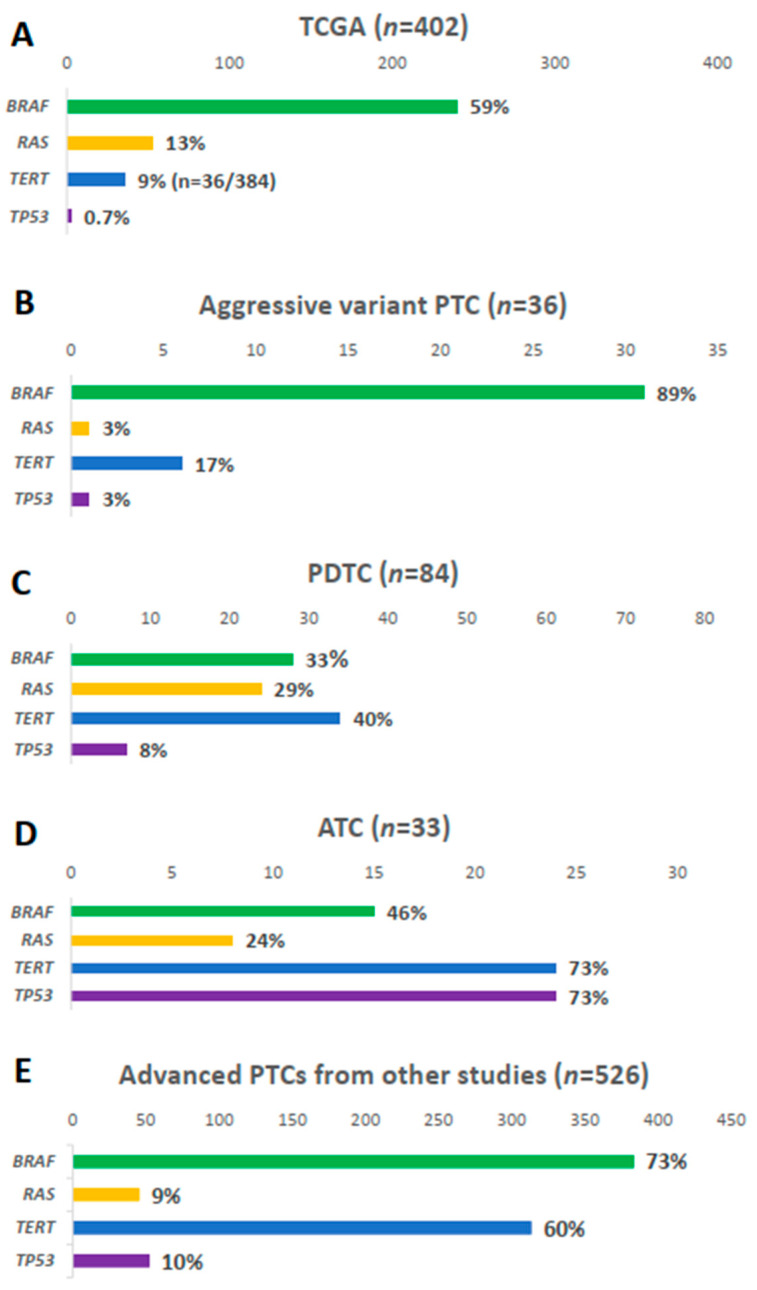
Comparison of *BRAF, RAS, TERT* promoter, and *TP53* mutations: (**A**) papillary thyroid carcinoma (PTC) from The Cancer Genome Atlas (TCGA); (**B**) aggressive variants of PTC in current study; (**C**) poorly differentiated carcinoma (PDTC) and (**D**) anaplastic thyroid carcinoma (ATC) from the Memorial Sloan Kettering Cancer Center (MSKCC) group; (**E**) advanced PTCs from other studies [17,19,20,21].

**Figure 3 cancers-13-00892-f003:**
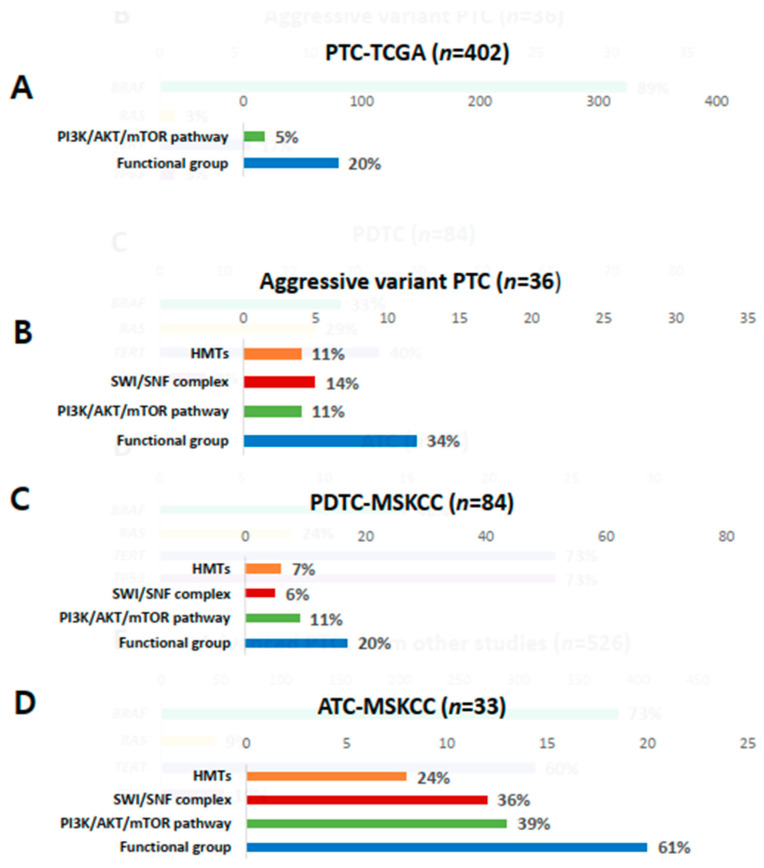
Comparison of mutations in histone methyltransferases (HMTs), switch/sucrose non-fermenting (SWI/SNF), the PI3K/AKT/mTOR pathway, and total functional groups. (**A**) Papillary thyroid carcinoma (PTC) from the Cancer Genome Atlas (TCGA); (**B**) aggressive variants of PTC; (**C**) poorly differentiated carcinoma (PDTC) from the Memorial Sloan Kettering Cancer Center (MSKCC) group; (**D**) anaplastic thyroid carcinoma (ATC) from the MSKCC group.

**Table 1 cancers-13-00892-t001:** Baseline clinical and pathological characteristics of patients with aggressive variants of papillary thyroid carcinoma (PTC).

Characteristics	Number (%) or Median (IQR)
Gender, Female	27 (75.0%)
Age (years)	43.5 (34.8–51.0)
≥55 years	7 (19.4%)
Pathology	
TCV-PTC	25 (69.4%)
CCV-PTC	11 (30.6%)
Surgery	
Lobectomy	2 (5.6%)
TT with CND	16 (44.4%)
TT with MRND	18 (50.0%)
Primary tumor size	2.0 (1.2–2.8)
>2 cm	17 (47.2%)
Extrathyroidal extension	
Microscopic	10 (27.8%)
Gross	19 (52.8%)
Multifocality (yes)	17 (47.2%)
Cervical lymph node metastasis (yes)	32 (88.9%)
N1a	14 (43.8%)
N1b	18 (56.2%)
Distant metastasis	0 (0.0%)
AJCC TNM 8th stage	
Stage I	31 (86.1%)
Stage II	4 (11.1%)
Stage III	1 (2.8%)
Stage IV	0 (0.0%)

Abbreviations: IQR, interquartile range; TCV, tall cell variant; PTC, papillary thyroid carcinoma; CCV, columnal cell variant; TT, total thyroidectomy; CND, central neck dissection; SND, selective neck dissection; MRND, modified radical neck dissection; AJCC, American Joint Committee on Cancer; TNM, tumor–node–metastasis.

**Table 2 cancers-13-00892-t002:** Comparison of mutational profiles in pathological subtypes of thyroid carcinoma.

Pathological Subtypes of Thyroid Cancer	PTC-TCGA (*n* = 402)	Aggressive Variant PTC (*n* = 36)	PDTC-MSKCC (*n* = 84)	ATC-MSKCC (*n* = 33)
Key pathways and functional groups	20%	34%	20%	61%
HMTs	-	11%	7%	24%
*KMT2A*	1%	0%	4%	9%
*KMT2C*	1%	11%	2%	3%
*KMT2D*	0%	-	1%	15%
*SETD2*	0.2%	-	0%	3%
SWI/SNF complex	-	14%	6%	36%
*ARID1A*	0%	0%	1%	9%
*ARID1B*	1%	6%	4%	9%
*ARID2*	0.2%	6%	1%	3%
*ARID5B*	0.2%	0%	0%	3%
*SMARCB1*	0%	0%	0%	6%
*PBRM1*	0%	0%	1%	6%
*ATRX*	0.2%	3%	0%	6%
PI3K/AKT/mTOR pathway	5%	11%	11%	39%
*PIK3CA*	0.4%	0%	2%	18%
*PTEN*	1%	0%	4%	15%
*PIK3C2G*	0.2%	0%	1%	6%
*PIK3CG*	0.2%	0%	1%	6%
*PIK3C3*	0%	-	1%	0%
*PIK3R1*	0.2%	-	1%	0%
*PIK3R2*	0.2%	-	0%	3%
*AKT1*	0.6%	3%	0%	0%
*AKT3*	1%	3%	1%	0%
*TSC1*	0.8%	-	1%	0%
*TSC2*	0.2%	0%	0%	3%
*MTOR*	0%	0%	1%	6%

Abbreviations: TCGA, The Cancer Genome Atlas; PTC: papillary thyroid carcinoma; PDTC, poor differentiated thyroid carcinoma; ATC, anaplastic thyroid carcinoma; HMTs, histone methyltransferase; SWI/SNF, switch/sucrose nonfermenting; PI3K, phosphoinositide 3-kinase; AKT, protein kinase B(PKB/AKT); mTOR, mammalian target of rapamycin.

## Data Availability

The data presented in this study are available in the present manuscript or in the Appendix A.

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
