# Peer review of "Genetic Profiles of Aggressive Variants of Papillary Thyroid Carcinomas"

_cancers, 2021, doi:10.3390/cancers13040892_

Round 1

Reviewer 1 Report

A few points:

-“These variants often display varying degrees of aggressive behavior in the spectrum

between well-differentiated, classic PTC and undifferentiated, anaplastic carcinoma”

-This sentence sounds like these variants are kind of histological dedifferentiation markers in the way from the good (classical PTC) to bad (anaplastic TC). There is no solid evidence for that. Please rephrase it accordingly. Other point please use either anaplastic or undifferentiated, they are the same.

-“The current American Thyroid Association (ATA) guidelines define aggressive variants of PTC as the following three subtypes, tall cell variant (TCV), columnal cell variant (CCV), and hobnail variant (HV)”

-In ICCR (International Collaboration on Cancer Reporting) also points which variants are aggressive in PTC which are different from ATA. Please mention, although not validated, there are other possible aggressive variants of PTC.

-The study shows 32 cervical LN metastasis. How many of them were central how many of them were lateral? Can authors mention about these a bit more? The surgical protocols are different in this manner, it would be great if the authors also would compare genetical profile in this aspect.

-please check the minor spelling errors

Author Response

Response to Reviewer 1

We appreciate your review of our manuscript. We believe that your comments have helped us to improve our manuscript. In the revised manuscript, changes are shown in bold red text.

  1. These variants often display varying degrees of aggressive behavior in the spectrum between well-differentiated, classic PTC and undifferentiated, anaplastic carcinoma” This sentence sounds like these variants are kind of histological dedifferentiation markers in the way from the good (classical PTC) to bad (anaplastic PTC). There is no solid evidence for that. Please rephrase it accordingly. Other point please use either anaplastic or undifferentiated, they are the same.

Thank you for your comment. We have rephrased this part (Page 2, Line 49-50).

  1. “The current American Thyroid Association (ATA) guidelines define aggressive variants of PTC as the following three subtypes, tall cell variant (TCV), columnal cell variant (CCV), and hobnail variant (HV)”In ICCR (International Collaboration on Cancer Reporting) also points which variants are aggressive in PTC which are different from ATA. Please mention, although not validated, there are other possible aggressive variants of PTC.

→ We have added the description at the introduction part. (Page 2, Line 50-53).

  1. The study shows 32 cervical LN metastasis. How many of them were central how many of them were lateral? Can authors mention about these a bit more? The surgical protocols are different in this manner, it would be great if the authors also would compare genetical profile in this aspect.

→ Among the 32 patients who had cervical LN metastasis, 14 patients had central neck LN metastasis, and other 18 had lateral LN metastasis. We have revised the Table1. (Page 2, Line 50-53). We have compared the genetic profiles of N1a and N1b tumors, but there was no significant difference in all genes between two groups (data not shown in manuscript).

  1. please check the minor spelling errors

The typing errors have been revised accordingly. Sorry for your inconvenience.

Reviewer 2 Report

The authors have analyzed the genetic profiles of aggressive variants of PTC by targeted sequencing. They have compared the prevalence of mutations in these variants with the PTC samples from the TCGA database and PDTC/ATC samples from the MSKCC dataset. Overall, it is a nice study and a good addition to the existing knowledge.

The only suggestion to the authors is to create a table enlisting the analyzed mutations within the genes. It could be added as a supplemental table.

Page 5, Line 160: The percentage of BRAF mutation in TCV-PTC and CCV-PTC is incorrect.

Page 1, Line 38: The authors should add "In conclusion" before "Aggressive variants of PTC...." to indicate that the last sentences are the concluding statements.

Author Response

Response to Reviewer 2

We appreciate your review of our manuscript. We believe that your comments have helped us to improve our manuscript. In the revised manuscript, changes are shown in bold red text.

The authors have analyzed the genetic profiles of aggressive variants of PTC by targeted sequencing. They have compared the prevalence of mutations in these variants with the PTC samples from the TCGA database and PDTC/ATC samples from the MSKCC dataset. Overall, it is a nice study and a good addition to the existing knowledge.

  1. The only suggestion to the authors is to create a table enlisting the analyzed mutations within the genes. It could be added as a supplemental table.

→ We have added the list of 50 target genes at the supplementary Table1.

  1. Page 5, Line 160: The percentage of BRAF mutation in TCV-PTC and CCV-PTC is incorrect.

→ We have revised the percentage accordingly. (Page 5, Line 164-165). Sorry for your inconvenience.

  1. Page 1, Line 38: The authors should add "In conclusion" before "Aggressive variants of PTC...." to indicate that the last sentences are the concluding statements.

Thank you for your comment. We have added “In conclusion” in accordance with your suggestion. (Page 1, Line 38-39)

Reviewer 3 Report

Jin et al. aimed to investigate the genetic profiles of aggressive variants of papillary thyroid cancer (PTC) by targeted next generation sequencing (NGS) and compare them with PTC from The Cancer Genome Atlas project and poorly differentiated thyroid cancers (PDTCs)/anaplastic thyroid cancers (ATCs) from the Memorial Sloan Kettering Cancer Center.They found out that aggressive variants PTC had higher prevalence in BRAF mutation (89%), and lower prevalence in NRAS mutation (3%) than other thyroid cancer. The mutation in TERT promoter (17%), tumor suppressor genes (23%), and genes encoding three functional groups (HMTs, SWI/SNF chromatin remodelling complex, and the PI3K/AKT/mTOR pathway) ranged between that of PTCs from TCGA and PDTCs and ATC from the MSKCC group. These genetic profiles might be associated with clinical outcomes of aggressive variants of PTC. The study is well conducted and the results of the study are very important contribution to the knowledge of thyroid cancer. Furthermore, their study is clearly written and tables and figures are very illustrative. My view is that paper of Jin et al. should be published in Cancers Journal. However, some editorial issues that the authors should be correct before the publication of their manuscript.

Editorial remarks:

  • From the line 98 on reference numbers are not cited sequentially.
  • Reference in line 111 is missing
  • Line 160: 2392.0% (probably 92%???)
  • Line 160: 872.7% (probably 72.2%???)

Author Response

Response to Reviewer 3

We appreciate your review of our manuscript. We believe that your comments have helped us to improve our manuscript. In the revised manuscript, changes are shown in bold red text.

Jin et al. aimed to investigate the genetic profiles of aggressive variants of papillary thyroid cancer (PTC) by targeted next generation sequencing (NGS) and compare them with PTC from The Cancer Genome Atlas project and poorly differentiated thyroid cancers (PDTCs)/anaplastic thyroid cancers (ATCs) from the Memorial Sloan Kettering Cancer Center. They found out that aggressive variants PTC had higher prevalence in BRAF mutation (89%), and lower prevalence in NRAS mutation (3%) than other thyroid cancer. The mutation in TERT promoter (17%), tumor suppressor genes (23%), and genes encoding three functional groups (HMTs, SWI/SNF chromatin remodeling complex, and the PI3K/AKT/mTOR pathway) ranged between that of PTCs from TCGA and PDTCs and ATC from the MSKCC group. These genetic profiles might be associated with clinical outcomes of aggressive variants of PTC. The study is well conducted and the results of the study are very important contribution to the knowledge of thyroid cancer. Furthermore, their study is clearly written and tables and figures are very illustrative. My view is that paper of Jin et al. should be published in Cancers Journal. However, some editorial issues that the authors should be correct before the publication of their manuscript.

Editorial remarks:

  1. From the line 98 on reference numbers are not cited sequentially.

Reference in line 111 is missing

Thank you for your comment. We have revised the reference accordingly.

  1. Line 160: 2392.0% (probably 92%???) Line 160: 872.7% (probably 72.2%???)

→ We have revised the percentage accordingly. (Page 5, Line 164-165). Sorry for your inconvenience.